# The Effects of Endogenous Hormones on the Flowering and Fruiting of *Glycyrrhiza uralensis*

**DOI:** 10.3390/plants8110519

**Published:** 2019-11-17

**Authors:** Binbin Yan, Junling Hou, Jie Cui, Chao He, Wenbin Li, Xiaoyu Chen, Min Li, Wenquan Wang

**Affiliations:** 1Chengdu University of Traditional Chinese Medicine, Chengdu 610075, China; yb51598@126.com; 2Institute of Medicinal Plant Development, Chinese Academy of Medical Sciences & Peking Union Medical College, Beijing 100193, China; jcui@implad.ac.cn (J.C.); hc891215@126.com (C.H.); cxy1033@163.com (X.C.); 3School of Chinese Pharmacy, Beijing University of Chinese Medicine, Beijing 100102, China; 602036@bucm.edu.cn (J.H.); wbli92@126.com (W.L.)

**Keywords:** *Glycyrrhiza uralensis* Fisch., flowering, fruiting, gibberellic acid, abscisic acid, zeatin riboside, indoleacetic acid

## Abstract

Although endogenous hormones play an important role in flower bud differentiation and seed-filling, their effects on the flowering and fruiting of *Glycyrrhiza uralensis* Fisch. remain unknown. In the present study, we investigate the differences in the levels of endogenous hormones gibberellic acid (GA), abscisic acid (ABA), zeatin riboside (ZR), and indoleacetic acid (IAA) between the fruiting and seedless plants of *G. uralensis* Fisch. at different growth stages. We also determine the correlations of the endogenous hormone with the rates of flower and fruit falling, rate of empty seeds, rate of shrunken grains, and thousand kernel weight (TKW). The results demonstrate that the IAA and ZR levels of the flowering plants are significantly higher than those of the nonflowering plants at the flower bud differentiation stage. The GA and ABA levels of exfoliated inflorescence plants are considerably higher than those of the flowering and fruiting plants; the rates of falling flowers and fruit are negatively correlated with the IAA level and positively correlated with the ABA level. The ABA content of nonflowering plants is significantly higher than that of fruiting plants. The ZR:GA and IAA:ABA ratios are significantly positively correlated with TKW. The IAA:GA and IAA:ABA ratios are significantly negatively correlated with the rates of empty and shrunken seeds. Thus, we speculate that high IAA and ZR contents are good for flower bud differentiation and seed-filling, and low ABA and ZR contents are beneficial to flower bud development and seed-filling.

## 1. Introduction

*Glycyrrhiza uralensis* Fisch., a perennial herb of the *Leguminosae* family [1], is a common bulk herb in China and has been widely used in the medical, food, tobacco, fodder, and cosmetic industries [2,3,4]. Wild *G. uralensis* has been the main source of licorice for decades. Overharvesting has gradually exhausted wild resources, and cultivated *G. uralensis* has become an alternative source. Although *G. uralensis* Fisch. exhibits seed propagation, its seed yield is low and its fruiting rate under natural conditions is only 10–21% [5]. Thus, the current *G. uralensis* production fails to satisfy the industrial demand. Flower bud differentiation and seed-filling are essential in the fruiting process of plants. Endogenous hormones play an important role in these processes. However, the effects of endogenous hormones on the flowering and fruiting of *G. uralensis* Fisch. are unknown.

Flower bud differentiation is an important sign that a plant is undergoing a transition from vegetative growth to reproductive growth [6]. This transition covers many complicated morphological and physiological changes and is the comprehensive response of plants to various signal states [7]. Endogenous hormones are the key factors in flower bud differentiation [8,9,10,11,12] and are important media for the formation of flower organs [13,14,15]. The plant hormones that considerably influence flower bud differentiation are gibberellic acid (GA), abscisic acid (ABA), zeatin riboside (ZR), and indoleacetic acid (IAA) [16]. Their effects on flowering vary between plants [17,18]. For example, GA can facilitate flower formation in plants, such as long-day (LD) and biennial plants [19], and inhibits it in others, such as apples [20], lychees [21], and *Arabidopsis thaliana* [22]. However, flower bud differentiation is a very complex process, and a single hormone by itself is not enough to have a significant impact on plants [7,23]. In roses, flower bud differentiation is greatly influenced by the hormone content and ratio because they affect the use of nutrients. High ABA:GA3, ABA:IAA, ZR:GA3, and ZR:IAA ratios are conducive to flower bud differentiation[18]. In particular, a high ZR:GA ratio is beneficial to apple bud formation [24], and a high ABA:GA ratio is conducive to flower bud initiation and differentiation in *Lycoris radiata* [25]. In cotton, high ZR:IAA and GA_3_:IAA imply a high number of flower buds[26].

Plant fruiting is continuously coordinated by plant hormones [27]. ABA, which instigates embryonic development [28], regulates seed-filling. The ABA content is positively correlated with the maximum seed-filling rate and maximum seed weight of wheat and corn [29,30]. The GA content is high in poor-quality wheat and corn seeds, but low in high-quality seeds [29,31]. IAA and ZR favor ovule development [32]. The combined effects of a pair of hormones on seed development are more crucial than the effects of a single hormone and depend on the ratio between the hormones [33]. The (IAA+GA+ZT): ABA ratio in lychees and jujube changes when the ABA content sharply increases, causing embryo development failure [34,35]. High ABA and Z+ZR contents and a low GA content considerably increase the accumulation of dry substances in seeds, thereby increasing corn and wheat seed yields [33,34].

This study aims to explore the relationship between endogenous hormones and the flowering and fruiting of *G. uralensis* by comparing the differences in endogenous hormone content in flowering fruiting, and nonflowering licorice plants at different growth stages and different ages. Furthermore, the dynamic changes of the endogenous hormones at different growth stages, the correlations of the endogenous hormones with the number of inflorescences, the rates of flower and fruit falling, and the rate of fruiting are analyzed. This study not only provides theoretical references for interpreting the flowering mechanism of *G. uralensis* Fisch., but also lays a theoretical foundation for the control of seed yields.

## 2. Results

### 2.1. Changes in Endogenous Hormones during the Growth of G. uralensis Fisch.

By comparing the changes in the endogenous hormones in two-, three-, and seven-year-old plants at different stages, we analyzed the changes in the endogenous hormones during plant growth. The results (Figure 1) showed that the changes in the plant hormones were the same in different years. The GA level was the highest (19.78 ng/g) at the germination stage, but the high levels only happened for seven-year old plants while two- and three-year old plants contained much less GAs and that this content did not change during time. At the vigorous-growth stage of the aboveground part, GA levels decreased to 4.36–5.11 ng/g and at the growth-termination stage of the aboveground part GA levels gradually increased (Figure 1A). The ZR level was low in the germination stage (6.64–10.33 ng/g) but gradually increased with vigorous life activities. The ZR level was the highest in the summer (11.06–16.47 ng/g). It increased during the first 2 growing months by 51%, 59%, and 67% in two-, three-, and seven-year old plants, respectively. At the end stage of aboveground growth, the ZR level gradually decreased. The ZR levels of two- and three-year-old plants were significantly higher than those of seven-year-old plants at different stages, but no remarkable difference was observed between two- and three-year-old plants (Figure 1B). The variation in the tendency of IAA levels was the same as in ZR. At the germination stage IAA levels were low and then increased gradually with vigorous life activities. In summer, the IAA content was the highest, and then decreased gradually at the end of growth stage of the aboveground part (Figure 1C). ABA level was low at the germination stage (53.64–61.89 ng/g) and gradually increased at the early growth stage. No significant change was observed in the vigorous-growth stage of the aboveground part. The content of ABA increased gradually from the vigorous-growth period to the stop growth period of the aboveground part (107.14–116.08 ng/g) (Figure 1D).

The samples were taken in field conditions to allow differences in weather conditions (wind, rain, temperature, etc.) to impact hormone levels in experiments where different time points were analyzed.

The ratio between growth-promoting hormones (ZR and IAA) and GA gradually increased from the spring germination stage to the summer vigorous-growth stage. Two-, three-, and seven-year-old plants increased by 126%, 109%, and 403%, respectively, and then gradually decreased at the end of autumn growth (Figure 2A). The ratio between growth-promoting hormones (ZR and IAA) and ABA remained at a high level (0.70–0.79; Figure 2B). This ratio rapidly decreased in autumn when growth stopped, decreasing by 38%, 37%, and 34% at two, three, and seven years, respectively. The ratio of GA to ABA rapidly decreased from the germination stage to the vigorous-growth stage, decreasing by 38%, 37%, and 34% at two, three, and seven years, respectively, and remained low until autumn, when growth stopped (Figure 2C).

The highest GA level was obtained in the germination stage, the highest IAA and ZR levels were obtained in the vigorous-growth stage, and the highest ABA level was obtained in the aboveground part of the growth-termination stage.

### 2.2. Changes in ZR and IAA Levels during Flower Bud Differentiation of G. uralensis

The effects of the endogenous hormones on the flower bud differentiation of licorice were analyzed by comparing the differences in the levels of the endogenous hormones among five-, seven-, and 12-year-old flowering and fruiting plants and nonflowering plants during flower bud incubation (5/11) and the bud stage (5/21). The results (Figure 3) showed no considerable difference between the ABA and GA levels of the flowering and nonflowering plants. The IAA and ZR levels of the flowering plants were significantly higher than those of the nonflowering plants (*p* < 0.05).

All the above data indicated that ZR and IAA may be related to flower bud differentiation of *G. uralensis*. High ZR and IAA levels may be benefited flower bud differentiation in *G. uralensis*.

### 2.3. Differences of the Endogenous Hormones between Fruiting and Nonflowering Plants at Different Stages

The 100 selected five-, seven-, and 12-year-old plants covered both fruiting and nonflowering plants (Appendix A). In Figure 4, the hormone contents of the flowering, fruiting plants, and nonflowering plants are the average hormone contents of the five-, seven-, and 12-year-old flowering and fruiting plants, as well as nonflowering plants. This study aimed to compare the differences in endogenous hormone levels among the flowering and fruiting plants and nonflowering plants at different stages. The IAA and ZR contents of the flowering and fruiting plants were significantly higher (*p* < 0.05) than those of the nonflowering plants at the flower bud differentiation stage (5/11). The IAA and ZR contents of the flowering and fruiting plants were substantially higher and lower than those of the nonflowering plants, respectively (*p* < 0.05), at the seed-filling stage (7/17). We speculated that many ZR participated in the seed-filling process at this time; hence, the ZR content in leaves was low. The ABA content in the nonflowering plants was significantly higher (by 48.27%) than that in the flowering and fruiting plants. No significant difference was observed in the endogenous hormone contents between flowering and nonflowering plants after seed maturation (9/13).

All the above data indicated that ZR and IAA may be related to the flowering and fruiting of *G. uralensis*. The high ZR and IAA levels may be beneficial to the flowering and fruiting processes of licorice, and a high level of ABA may be unfavorable to the fruiting of *Glycyrrhiza.*

### 2.4. Correlation between the Endogenous Hormones and Flower and Fruit Falling Rates

Flower and fruit falling covers two situations, namely inflorescence falling, which was observed in some plants in all sampling plots (Appendix A), and floret or pod falling (Appendix A). To explore the relationships between the endogenous hormones and flower and fruit falling, we analyzed the differences in endogenous hormone contents between the fruiting plants and inflorescence falling plants from the squaring stage to the seed-filling stage. Correlations between flower and fruit falling rates and the endogenous hormone contents were analyzed.

The results (Figure 5) showed that the GA and ABA levels of exfoliated inflorescence plants were considerably higher than those of the flowering and fruiting plants (*p* < 0.05); the IAA and ZR levels of flowering and fruiting plants were not significantly different from those of exfoliated inflorescence plants. The rates of falling flowers and fruit were negatively correlated with the IAA level at the budding and filling stages and positively correlated with the ABA level at the filling stage (Table 1).

The above results showed that there was a strong correlation between endogenous hormones and the flower and fruit falling of *Glycyrrhiza uralensis*. High GA and ABA levels are possible causes of licorice inflorescence shedding. High ABA and low IAA levels may be the cause of licorice flower and fruit loss.

### 2.5. Correlations of the Endogenous Hormone with the Number of Empty Seeds, Number of Shrunken Seeds, and TKW

The correlations of endogenous hormone content and ratio with the rate of empty seeds, rate of shrunken seeds, and TKW in different sampling plots (Appendix A) were analyzed (Table 2). The number of empty seeds, number of shrunken seeds, and TKW were remarkably correlated with the level of the endogenous hormones at the grain-filling stage, but not with the level of the endogenous hormones at the other stages. The rate of empty seeds was significantly negatively correlated with IAA content, IAA:GA ratio, and IAA:ABA on 7/17 (seed-filling stage). During the same period, the rate of shrunken seeds showed a significantly negative correlation with IAA:GA and IAA:ABA ratios. TKW was significantly positively correlated with ZR content and IAA:ABA ratio during the seed-filling stage, but exhibited a significantly positive correlation with ZR/GA ratio.

The above data showed that there was a strong correlation between endogenous hormones and the fructification of Glycyrrhiza uralensis. IAA and ZR may be beneficial to the seed setting of Glycyrrhiza, while GA and ABA may be unfavorable to the seed setting of Glycyrrhiza.

## 3. Discussion

The ABA plays an important role in plant growth and development. During the growth and development of different plants, the change trend of ABA content is almost the same. The ABA content of most plants will increase to a higher level in the later stage of growth or fruit maturity. In this study, ABA level was low at the germination stage and gradually increased at the early growth stage. No significant change was observed in the vigorous-growth stage of the aboveground part. The content of ABA increased gradually from the vigorous-growth period to the stop growth period of the aboveground part. This result is similar to the change trend of ABA content during the growth and development of raspberry (*Rubus idaeus* L.) [36], peach [37], spring maize [38], tea tree [39], wheat [40] and lycium [41]. The content of ABA in the growth process of raspberry (*Rubus idaeus* L.), peach and tea tree were lower in the early growth stage and vigorous-growth stage, and highest in the slow growth and rest stage. The ABA content of spring maize, lyceum, and wheat fruit increased gradually during the process from formation to maturity. The research shows that the ABA appears to modulate fruit ripening through interference not only with ethylene and the cell wall but also with auxin-related genes [42]. Excess biosynthesis of ABA may deprive the same precursor pool necessary for the chlorophyll biosynthesis pathway, thereby triggering growth retardation [43]. It is speculated that in the later stage of plant growth, a large amount of ABA is generated by the plant itself, which slows down its growth speed, promotes seed maturity, and causes the plant to enter the aging or dormancy period.

The effects of IAA, ZR, GA, and ABA on the flowering of different plants are different [17,18,44]. The IAA content in *Agapanthus praecox* spp. is increased by 581% between vegetative growth and flower bud formation. According to plant hormone immune localization analysis, IAA participates in the differentiation and development of each flower organ [13]. The IAA content in cotton is at a minimum at the beginning of flower bud differentiation [45]. In the present study, the IAA contents of five-, seven-, and 12-year fruiting plants during flower bud formation are significantly higher than those of seedless plants. The number of inflorescences shows a significantly positive correlation with IAA content during flower bud formation, indicating that IAA can facilitate the flower bud differentiation of *G. uralensis* Fisch. At a certain range, the number of inflorescences was high given the high IAA content. ZR can facilitate flower bud differentiation in most plants. The ZR content in roses reaches the highest level during the squaring stage [18]. In this study, the ZR content of the seven-year-old fruiting plants peaked during the squaring stage. The ZR contents of five-, seven-, and 12-year-old fruiting plants during the flower bud formation were significantly higher than those in seedless plants. The number of inflorescences showed a significantly positive correlation with ZR content. This reveals that ZR can facilitate flower bud differentiation in *G. uralensis* Fisch. A high ZR content in a certain range can cause a high number of inflorescences. This result is consistent with the effects of ZR content in *Arabidopsis*—that is, the ZR content increases dramatically during the flower bud formation of *Arabidopsis* and can facilitate its reproduction and meristem growth. The increased ZR in inflorescences generates additional meristem tissues and thereby produces additional flower primordia. Moreover, the increased ZR content increases the number of cells and thereby expands the floral organ, especially a pistil with ovule [46].

Flower bud differentiation is a very complicated process. All hormones can cause profound effects on flower bud differentiation. However, these hormones do not independently influence flowering but restrict and interact mutually. For example, ZR hinders GA production and facilitates GA degradation, whereas GA inhibits the ZR response [47]. An equilibrium state is produced by the interaction of different proportions of various hormones. Such equilibrium comprehensively influences flower bud differentiation in plants by influencing the metabolism of nutrients, such as nucleic acids, proteins, and soluble sugars. For instance, high IAA:GA, ZR:ABA, and IAA:ABA ratios are conducive to the flowering of ginkgo [48]. In the present study, the number of inflorescences has a significantly positive correlation with ZR:GA, IAA:GA, and IAA:ABA ratios during flower bud differentiation. This finding implies that high ZR:GA, IAA:GA, and IAA:ABA ratios favor flower bud differentiation in *G. uralensis* Fisch. The influences of single hormones on flower bud differentiation suggest that IAA and ZR play dominant roles.

Plant hormones play important regulatory roles in seed-filling. In the normally developed embryos of li-jujube and lychees, IAA, GA, and CTK (cytokinin) contents are high, but ABA content is low [27]. The high content and proportion of ABA results in the seed-filling failure of loquats [49]. In the present study, the ABA contents of the five-, seven-, and 12-year-old fruiting plants during the seed-filling stage was significantly lower than those of seedless plants, but no significant difference was observed during the maturation stage. This suggests that ABA can considerably inhibit the seed-filling of *G. uralensis* Fisch. Reed et al. discussed the relationship between changes in endogenous hormones and embryonic development failure in corn seeds; they concluded that changes in a single endogenous hormone are inadequate to explain the cause of embryonic development failure and that failed seeds contains a high ABA content and low IAA content [50]. The poor filling of lower spikelets in rice is mainly caused by the reduction of the ABA, IAA, and PA (polyamines) contents, which further causes a reduction in the cell division rate and albuminous cells, a deceleration of the filling rate, and a decrease in seed weight [51]. In the present study, the IAA:ABA ratio of the fruiting plants during the seed-filling stage was higher than the ratio of the seedless plants, indicating that the IAA:ABA ratio facilitate the seed-filling of *G. uralensis* Fisch.

Hormonal imbalance is another important cause of flower and fruit falling and failure of seed-filling. In the present study, the rate of flower and fruit falling showed significantly negative correlations with IAA and ABA contents, whereas the rate of empty or shrunken seeds exhibited a significantly negative correlation with IAA:GA and IAA:ABA ratios. Increases in TKW was significantly positively correlated with ZR content and ZR:GA and IAA:ABA. Based on this finding, increasing the ZR:GA, IAA:GA, and IAA:ABA ratios can decrease the rate of flower and fruit falling and increase the rate of fruit and TKW. The seed yield of *G. uralensis* Fisch. can be increased by increasing the IAA and GA contents or decreasing the ABA and GA contents. Exogenous GA 3 decreases the seed yield of *Carthamus tinctorius* [52]. The filling rate of corn seeds is increased when uniconazole is used, which increases the ZR and ABA contents and decreases GA content [53]. Uniconazole also increases the CK (cytokinin) content. In duckweed seeds, these features enable uniconazole to increase starch accumulation [54]. Meanwhile, the seed yield of *Arabidopsis* can be increased by 55% by adjusting the ZR content [46]. Therefore, future studies must investigate how the contents of endogenous hormones can be regulated for the increase of *G. uralensis* Fisch. seed yield. Currently, relevant experiments have been implemented.

## 4. Materials and Methods

### 4.1. Experimental Materials

*G. uralensis* Fisch. plants in different growth years were used. The test site is in Hedong Town, Guazhou County, Jiuquan City, Gansu Province (40°31′N, 96°42′E), where the altitude is approximately 1378 m. The mean annual precipitation is 83.5 mm. The maximum, minimum, and annual average temperatures are 36.7 °C, −17.6 °C, and 8.75 °C, respectively. The annual sunshine duration is 3237 h.

### 4.2. Experimental Design

Two-, three-, five-, seven-, and 12-year-old *G. uralensis* Fisch. plants were selected. Three sampling plots (5 × 5 m^2^) were selected randomly in the test field for the three repetitions. The study involved the following four experimental designs:

Two-, three-, and seven-year-old *G. uralensis* Fisch plants were considered to be subjects. In September 2015, 100 plants were stake-marked in quadrats of various ages. The bud of the plant labeled on 1 May, and the second compound leaves at the top of each labeled plant on 21 May, 17 July, and 13 September were selected and stored at −80 °C.

On 10 May 2016, 100 plants numbered 1–100 in the quadrat were listed and labeled as five-, seven-, and 12-year-old samples. The second compound leaf at the top of each marked plant was obtained from the budding stage (5/11), squaring stage (5/21), seed-filling stage (7/17), and seed maturation stage (9/13), and stored at −80 °C. After 13 September, the collected samples were grouped and merged according to flowering and fruiting conditions. Each growth year was divided into a flowering and fruiting group and a nonflowering group.

The sampling time for each period was between 8 a.m. and 9 a.m.

On 17 July 2016 (the filling stage), 2016, 100 plants were randomly labeled in the five-, seven-, and 12-year-old quadrats, and the flower and fruit abortion rates of each plant were counted. The empty seed rate and shrinkage rate of each plant were counted after 13 September, and the TKW of seeds in each sampling plot was tested at the same time.

### 4.3. Determination of the Endogenous Hormone Content

The concentrations of the endogenous hormones ZR, IAA, ABA, and GA were tested by enzyme-linked immunosorbent assay. Reagent kits were provided by the plant endogenous hormone laboratory of China Agricultural University [33,55]. Leaf samples (0.50 g) were collected and ground in liquid nitrogen. The powder was frozen, dried, and extracted for 4 h in 80% methyl alcohol (T = 4 °C). The mixture was centrifuged for 15 min at a rate of 4000 r/min, and the supernatant was collected, precipitated, and subjected to repeated extraction. The supernatants were mixed. The samples were precipitated in precooled 80% methyl alcohol through eddy and rotation and then washed three times. The supernatants of each sample were mixed and then dried in a Speedy Vac until approximately 50 µL of liquid remained. A TBS buffer solution containing 25 mL Tris-HCl (pH 7.5), 100 mL NaCl_2_, and 3 mL NaN_3_ were added until the 1.5 mL mark was reached. 

Extracts were diluted in TBS 10 times, and the standard curves of different diluted ZR, IAA, ABA, and GA solutions were constructed according to the reagent kit to test the endogenous hormone contents. The concentrations of plant hormones were calculated and expressed as ng/g. Each measurement was repeated three times.

### 4.4. Statistical Analysis

SPSS 19.0 was applied for data descriptive statistics, factor correlation analysis, and variance analysis (i.e., ANOVA). The contents of endogenous hormones in the flowering and nonflowering plants in each sample were determined three times, and variance analysis was performed using the average value of each sample. Data from each sampling date were analyzed separately, using the Duncan Analysis Method, and the resultant means were tested by the least significant difference test at the P_0.05_ level (LSD0.05). 

## 5. Conclusions

The content of endogenous hormones is closely related to the flowering and fruiting of G. uralensis Fisch. The results show that the IAA and ZR levels of flowering plants were significantly higher than those of nonflowering plants at the flower bud differentiation stage. The GA and ABA levels of exfoliated inflorescence plants were considerably higher than those of flowering and fruiting plants; the rates of falling flowers and fruit were negatively correlated with the IAA level at the budding and filling stages and positively correlated with the ABA level at the filling stage. It is suggested that endogenous hormones may be related to the flowering of *G. uralensis*. High IAA and ZR contents in the flower bud differentiation stage is conducive to flower bud differentiation. A high content of GA and ABA may lead to flower and fruit falling.

The ABA content of nonflowering plants was significantly higher than that of fruiting plants. The ZR:GA and IAA:ABA ratios are significantly positively correlated with TKW. The IAA:GA and IAA:ABA ratios are significantly negatively correlated with the rates of empty and shrunken seeds. The above data showed that there was a strong correlation between endogenous hormones and the fructification of Glycyrrhiza uralensis. IAA and ZR may be beneficial to the seed setting of Glycyrrhiza, while GA and ABA may be unfavorable to the seed setting of Glycyrrhiza.

Therefore, increasing IAA and ZR or decreasing ABA and GA may facilitate flower bud differentiation and seed-filling, increase the TKW, and decrease the rate of flower and fruit falling, empty seeds, and shrunken seeds. These effects ultimately increase the seed yield of G. uralensis Fisch.

## Figures and Tables

**Figure 1 plants-08-00519-f001:**
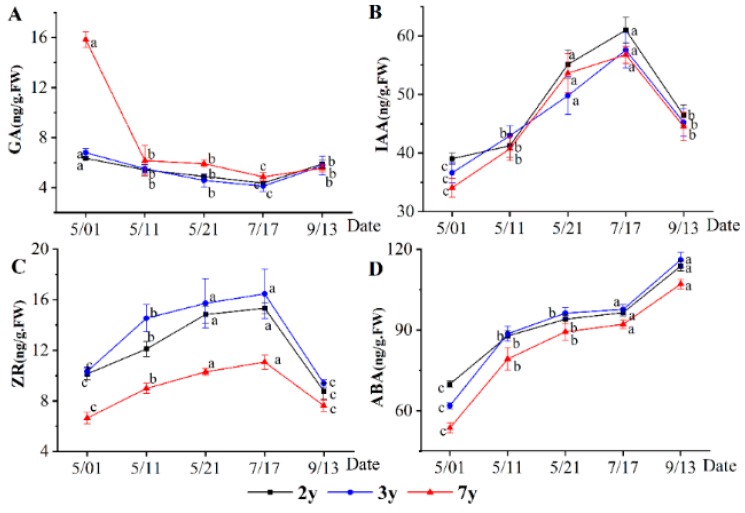
Changes in the endogenous hormones throughout the growth of *G. uralensis* Fisch. Different letters reflect significant differences among different indexes in the same stage: (**A**) GA content, (**B**) IAA content, (**C**) ZR content, and (**D**) ABA content.

**Figure 2 plants-08-00519-f002:**
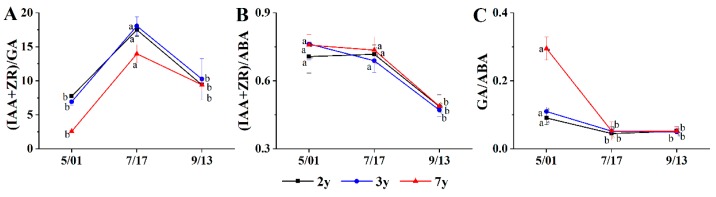
Changes in endogenous hormones ratios during the growth of *G. uralensis* Fisch. The different letters reflect significant differences among different indexes in the same stage e: (**A**) (IAA+ZR):GA, (**B**) (IAA+ZR):ABA, and (**C**) GA:ABA.

**Figure 3 plants-08-00519-f003:**
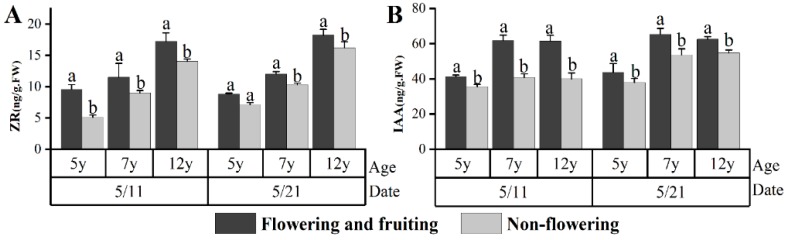
Effects of the endogenous hormone contents on *G. uralensis* Fisch. flower bud formation. Different letters reflect significant differences among different indexes in the same stage: (**A**) ZR content, (**B**) IAA content.

**Figure 4 plants-08-00519-f004:**
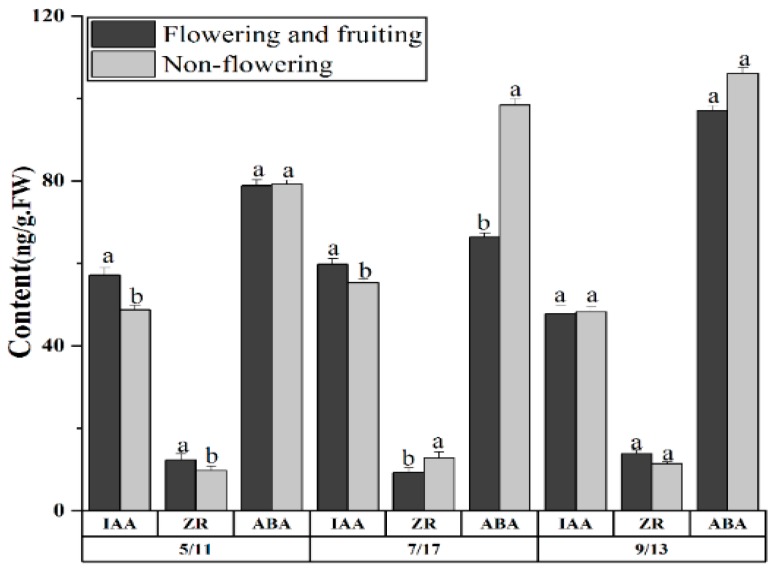
Differences in the endogenous hormones between flowering and fruiting plants and nonflowering plants at different growth stages. The different letters reflect significant differences among different indexes in the same stage.

**Figure 5 plants-08-00519-f005:**
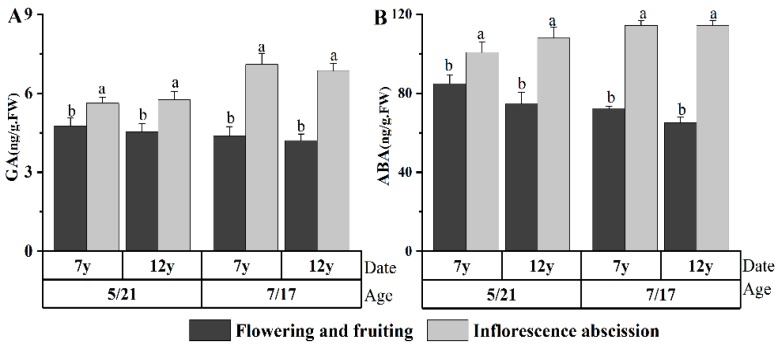
Correlations between the endogenous hormones and inflorescences’ abscission. The different letters reflect significant differences among different indexes in the same stage: (**A**) GA content and (**B**) ABA content.

**Table 1 plants-08-00519-t001:** Correlations between the endogenous hormones and inflorescence abscission.

Pearson Coefficient	Date	ABA	IAA
Rates of falling flowers and fruit	5/21	−0.231	−0.825 **
	7/17	0.805 **	−0.759 *

Note: * Means correlation at the 0.05 level (bilateral); ** Means correlation at the 0.01 level (bilateral).

**Table 2 plants-08-00519-t002:** Correlations of the endogenous hormones with rate of empty seeds, rate of shrunken seeds, and TKW (*n* = 9).

Pearson Coefficient	Date	IAA	ZR	ZR:GA	IAA:GA	IAA:ABA
Rate of empty seeds	7/17	−0.786 *	−0.105	−0.198	−0.701 *	−0.782 *
Rate of shrunken seeds	7/17	−0.631	−0.415	−0.569	−0.667 *	−0.789 *
TKW	7/17	0.527	0.773 *	0.839 **	0.482	0.675 *

Note: * Means correlation at the 0.05 level (bilateral); ** Means correlation at the 0.01 level (bilateral).

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
