# Peer review of "The Effects of Endogenous Hormones on the Flowering and Fruiting of Glycyrrhiza uralensis"

_plants, 2019, doi:10.3390/plants8110519_

Round 1
Reviewer 1 Report
The article "Effects of Endogenous Hormones on the Flowering and Fruiting of Glycyrrhiza uralensis" reports new findings worthy of publication in Plants. In this manuscript, the authors investigated the difference in the levels of endogenous hormones gibberellic acid (GA), abscisic acid (ABA), zeatin riboside (ZR), and indoleacetic acid (IAA) between the fruiting and nonfruiting plants of G. uralensis Fisch. at different growth stages. Determination of the correlations of the endogenous hormone with the number of inflorescences, rates of flower and fruit falling, rate of empty seeds, rate of shrunken grains, and thousand seed weight (TKW). The used methods were adequate. The data supports the results, which are properly discussed. Specifically, they found that GA is beneficial to flower bud formation in G. uralensis Fisch., and high IAA and ZR contents are good for flower bud differentiation and seed filling. Therefore, I recommend the manuscript for publication.
Author Response
Dear Reviewer:
Thank you for your comments concerning our manuscript entitled “The Effects of Endogenous Hormones on the Flowering and Fruiting of Glycyrrhiza uralensis” (ID: 632569). Those comments are all valuable and very helpful for revising and improving our paper, as well as the important guiding significance to our researches.
As for the language problem you pointed out, we invited experts from MDPI to edit the manuscript(Expert:Martyn Rittman, Ph.D. English Editing Manager. Certificate number:13604).Revised portion are marked in red in the paper.
Please see the attachment.
Once again, thank you very much for your comments and suggestions.

Reviewer 2 Report
plants-632569
Review Report
Title: Effects of Endogenous Hormones on the Flowering and Fruiting of Glycyrrhiza uralensis
The manuscript has been improved and the paper itself is well written.
The only suggestion I have is related to the section: Conclusion. Is a poor section, in my opinion, I recommend the authors to give and discuss in detail this section.
Also at •Line 291: “bud differentiation stage is conducive” instead of “bud differentiation stage are conducive to flower bud differentiation”.
Line 297:” contents facilitate flower bud differentiation” instead of “contents facilitates flower bud differentiation and seed filling”.
Author Response
Dear Reviewer:
Thank you for your comments concerning our manuscript entitled “The Effects of Endogenous Hormones on the Flowering and Fruiting of Glycyrrhiza uralensis” (ID: 632569). Those comments are all valuable and very helpful for revising and improving our paper, as well as the important guiding significance to our researches.
We have studied comments carefully and have made correction which we hope meet with approval. As for the language problem you pointed out, we invited experts from MDPI to edit the manuscript(Expert:Martyn Rittman, Ph.D. English Editing Manager. Certificate number:13604).Revised portion are marked in red in the paper.
Please see the attachment.
Once again, thank you very much for your comments and suggestions.

Reviewer 3 Report
In this manuscript Yan et al describe the hormone content of different tissues of Glycyrrhiza uralensis at different stages.
I am sorry to say that I have not been able to correctly review this manuscript due to very poor english language and style.
I am not myself a native english speaker but I think this manuscript requires extensive editing of English language and style. Sentences as the next examples were impossible to understand and interpret.
-The GA level was the highest(19.78 ng/g)at the germination stage, the vigorous-growth stage of the aboveground part decreased to 4.36–5.11 ng/g, and the growth-termination stage of the aboveground part gradually increased (Figure 1A.).
-The aboveground part growth termination stage gradually decreased.
-The variation in IAA level was the same as the variation in IAA level in ZR.
-The germination stage was low and then increased gradually with vigorous life activities. The peak-growth stage in summer was reached, and the growth-termination stage of the aboveground part was gradually reduced.
-The aboveground part increased during the growth-termination stage (107.14–116.08 ng/g).
Besides the english issue that can be easily fixed by the help of a colleague with better english skills or by using an english editing service, the authors should be very careful in discriminating between correlation of data and cause-effect relationship. Higher levels of a hormone in a flowering plant and low levels in a non flowering plant does not mean that this hormone helps with flowering. There is not a cause-effect. There is only a correlation.
The authors have gathered a lot of data that I will be very happy to review in a more carefully writed manuscript.
Author Response

(The authors gave the same response as above.)

Round 2
Reviewer 3 Report
The current version of the manuscript is easier to read.
I have some comments for the authors:
Fig 1 shows that ABA content gradually increases during development. Is this common to other plant species? Could you include some lines discussing this result in the discussion section?
Hormone levels are responsive to weather conditions. Since the samples were taken in field conditions. Could the authors mention somewhere that differences in weather conditions (windy, rainy, temnperature, etc) could also impac hormone levels in experiments where different time points are analized?
The authors state that “The GA level was the highest (19.78 ng/g) at the germination stage” but
this is true only for the 7 years-old plants. Please describe results properly and indicate that the high levels only happened for 7 year-old plants while 2 and 3-year old plants contained much less GAs and that this content did not change during time.
Some English corrections:
- The GA level was the highest (19.78 ng/g) at the germination stage. At the vigorous-growth stage of the aboveground part, GA levels decreased to 4.36–5.11 ng/g and at the growth-termination stage of the aboveground part GA levels gradually increased (Figure 1A).
- The ZR level was the highest in summer (11.06–16.47 ng/g). It increased during the first 2 growing months by 51%, 59%, and 67% in two-, three-, and seven-year old plants, respectively.
-Line 94. The variation in the tendency of IAA levels was the same as in ZR.
-Line 91. The aboveground part of the growth termination stage gradually decreased. What did decrease? The growth? The hormone? Please rewrite.
- The germination stage was low and then increased gradually with vigorous life activities. What was low? I think it is more correct as follows:
At The germination stage IAA levels were low and then increased gradually with vigorous life activities.
Please, correct the following sentences accordingly:
-The highest content in summer, and the growth-termination stage of the aboveground part was gradually reduced.
-2.2. “Effects of the endogenous hormones on the flower bud differentiation of G. uralensis” it is not correct. Please change by:
2.2. Changes on ZR and IAA levels during flower bud differentiation of G. uralensis

Author Response
Dear Reviewer:
Thank you for your comments concerning our manuscript entitled “The Effects of Endogenous Hormones on the Flowering and Fruiting of Glycyrrhiza uralensis” (ID: 632569). You read the manuscript very carefully and pointed out many very good problems.
We have studied comments carefully and have made correction which we hope meet with approval. Revised portion are marked in red in the paper. By correcting these problems, we have learned a lot, which is very helpful for our future research and language writing.
Please see the attachment.
Once again, thank you very much for your comments and suggestions.

This manuscript is a resubmission of an earlier submission. The following is a list of the peer review reports and author responses from that submission.
Round 1
Reviewer 1 Report
The manuscript by Yan et al submitted for publication in Plants aims to investigate the effect of plant hormones on flowering and fruiting of Glycyrrhiza uralensis. For the several reasons listed below I believe that this manuscript is not suitable for publication in Plants. Below I list some of the most relevant points.
· In the last paragraph the authors write “This study aims to investigate the effects of endogenous hormones on the flowering and fruiting of G. uralensis by comparing differences in endogenous hormone content…”. The authors do not provide any real functional data supporting that the investigated hormones have an effect on flowering (in fact flower morphogenesis) and fruiting but rather simple hormone measurements. This observation, by no mean, implies a causal effect in these processes.
· The authors seem to not be familiar with the differences between flowering transition (or flowering time) and flower development, which are regulated by completely different genetic pathways both in annual and perennial species. This is clearly noticed along the manuscript where both topics are mixed. This is further worsened in the discussion; authors have a full paragraph citing flowering time genes, no flower development. Finally, if the authors want to work on flowering time, there is a vast recent literature on how GA-DELLA-mediated flowering (time) pathway, which was not mentioned. Please differentiate both processes.
· I had a hard time to understand the correlation between hormone level and number of inflorescence. The transition to inflorescence meristem (inflorescence meristem) precedes the flower meristem transition. If I understood correctly, authors harvested flower buds for hormone measurement; therefore it should reflect the hormone level in flowers. How could the IAA and ZR in flower buds correlates with the number of inflorescence? I think I am missing some important information, please clarify.
· The authors do no provide any information on how hormones regulate flowers patterning; saying that it is complex or complicate does not help the reader to convey all the information provided and value the manuscript.
· There are wrong citations along the manuscript as well as many very important references missing. Retracted manuscripts must not be included as a reference (ref 12).
· Give credit to original publications instead of reviews.
· It is fundamental providing the exact material harvested in the material and methods. Hormones are dynamically regulating flower development; therefore their levels are not fixed over time. Moreover, the time of the day (or approximate ZT) samples were harvested should be provided as hormones are gated by circadian clock.
· Authors should look for professional English revision, the manuscript is poorly written.
Author Response
Dear Reviewer:
Thank you for your comments concerning our manuscript entitled “Effects of Endogenous Hormone on the Flowering and Fruiting of Glycyrrhiza uralensis” (ID: 555265). Those comments are all valuable and very helpful for revising and improving our paper, as well as the important guiding significance to our researches. We have studied comments carefully and have made correction which we hope meet with approval. Revised portion are marked in red in the paper.
Please see the attachment.
Because of the heavy workload of testing data determination and sampling in the field recently. Therefore, some questions may not be perfectly answered. If there are any questions, please point them out and I will timely correct them.
Once again, thank you very much for your comments and suggestions.

Reviewer 2 Report
The manuscript entitled "Effects of Endogenous Hormone on the Flowering and Fruiting of Glycyrrhiza uralensis" is interesting in view of the contents of endogenous hormones can significantly influence the flowering and fruiting of G. uralensis Fisch. High GA content and low ABA, ZR, and IAA contents during the germination stage are beneficial to flower bud formation. The paper described GA is beneficial to flower bud formation in G. uralensis Fisch., and high IAA and ZR contents are good for flower bud differentiation and seed filling. It is carefully done and well written and the results are new. The information generated are of major interest for the readers of Plants. I recommend publication of this interesting paper in Plants.
Author Response
Dear Reviewer:
Special thanks to you for your good comments. Your encouragement makes us more confident in our research work, and we hope that we can make better results to show you in the future.
Reviewer 3 Report
Review Report plants-555265
Article title: Effects of Endogenous Hormone on the Flowering and Fruiting of Glycyrrhiza uralensis
This manuscript describes an interesting work which investigates the difference in the levels of endogenous hormones GA, ABA, ZR and IAA between the fruiting and seedless plants of G. uralensis Fisch at different growth stages. This is an applicative idea and the manuscript is well presented.
Results demonstrate that high GA content and low ABA, ZR, and IAA contents during the germination stage are beneficial to flower bud formation and high IAA and ZR contents in the flower bud differentiation stage are conducive to flower bud differentiation. GA is beneficial to flower bud formation in G. uralensis Fisch., and high IAA and ZR contents are good for flower bud differentiation and seed filling.
However, there are some information are missing and it is not suitable to be published in its present form. Following revisions are suggested:
Line 17: I recommend replacing the ” nonfruiting” with “seedless”
Line 18: “the endogenous” instead of ‘’ endogenous”, “inflorescences” instead of “inflorescence”
Line 36: “are” instead of “is” and so on....
Line 43-44 ’’Endogenous hormones are the key factors in flower bud differentiation [8] and important media for the formation of flower organs [9]… please introduce also the recent references : Ethylene Measurements from Sweet Fruits Flowers Using Photoacoustic Spectroscopy, Molecules, 2019 Mar; 24(6): 1144. doi: 10.3390/molecules24061144; and Role of ethylene in responses of plants to nitrogen availability. Front. Plant Sci. 2015; 6:927. doi: 10.3389/fpls.2015.00927.
In general there are some grammatical mistakes, which should be carefully corrected. I suggest the author to spend some time to polishing the article and to carefully correct some grammatical mistakes.
The experimental method and statistical analysis is unclear. (Please give more details about ANOVA).
Author Response

(The authors gave the same response as above.)

Round 2
Reviewer 1 Report
Below a briefly list some points that authors should consider for improvements in this manuscript.
The hormone level in leaves is measured to correlate the amount of inflorescences. I am not such correlation is valid since there are several other factors affect branching and the number of inflorescences formed in a plant. Authors insist in mixing flowering induction, flower differentiation and fruiting. These are completely different biological processes in time and space. As mentioned in the last paragraph of the introduction, the aim of this study is to explore the correlation between the endogenous hormone with flowering, fruiting and non-flowering licorice plants in different ages and growth stage. The hormone levels inform more about the age and growth stage but there is no necessary correlation with number of inflorescence, rates of flower and fruit falling and rate of fruiting. Perhaps authors could indicate the full name of the plant LD in line 48. Also, in the next line: “flower bud differentiation is complicated process, and a single hormone influences plants”. What does it mean? I am really surprised that professional English revision was invited, perhaps authors could provide a certificate number for the language revivion. Authors insist in citing retracted manuscript (ref 45). The references 38-42 is neither the latest not the most important references on GA signaling on flowering transition. Also, they refer to very different biological processes (flowering transition, flower development and pollen development). The auxin and GA signaling on flower differentiation take place in very specific postion at the shoot meristem and SPL proteins mediate it. Authors should be familiar with the literature (eg. Yu et al., 2012 – Plant Cell, Hyun et al., 2016 – Developmental Cell, Yamaguchi et al., 2014 – Science). There is vast literature covering this topic, which was completely omitted.